# A Calculation Model for Cooling Rate of Aluminum Alloy Melts during Continuous Rheo-Extrusion

**DOI:** 10.3390/ma14195684

**Published:** 2021-09-29

**Authors:** Yu Wang, Minqiang Gao, Bowei Yang, Jingyuan Bai, Renguo Guan

**Affiliations:** 1Key Laboratory of Lightweight Structural Materials Liaoning Province, School of Materials Science and Engineering, Northeastern University, Shenyang 110819, China; neu_wangyu@yeah.net (Y.W.); ybw_neu@163.com (B.Y.); Baekkyungwon@163.com (J.B.); 2Engineering Research Center of Continuous Extrusion, Ministry of Education, Dalian Jiaotong University, Dalian 116028, China

**Keywords:** aluminum alloy, continuous rheo-extrusion, heat transfer, solidification, cooling rate

## Abstract

The melt temperature of aluminum alloys plays a significant role in determining the microstructure characteristic during continuous rheo-extrusion. However, it is difficult to measure the actual melt temperature in the roll-shoe gap. In this work, based on the basic theory of heat transfer, a calculation model for heat transfer coefficient of cooling water/roll interface and melt/roll interface is established. In addition, the relationship between the temperature at the melt/roll interface and the velocity of cooling water is investigated. Combined with the CALPHAD calculation, the melt temperature during solidification in the continuous rheo-extrusion process is calculated. Using this model, the cooling rate of an Al–6Mg (wt.%) alloy melt prepared by continuous rheo-extrusion is estimated to be 10.3 K/s. This model used to determine the melt parameters during solidification provides a reference for optimizing the production process of continuous rheo-extrusion technology.

## 1. Introduction

Green, energy-saving and efficient processing technology has been developed rapidly in recent decades [1,2,3]. Continuous rheo-extrusion technology is an advanced technology that can be directly used for continuous preparation of aluminum alloy billets and pipes and realize one-step forming from casting to finished products [4]. Compared with the traditional casting coupled with deformation processing technologies, the continuous rheo-extrusion technology possesses the advantage of reducing the capital cost, energy consumption, operation cost and scrap rate.

During the continuous rheo-extrusion process for preparing aluminum alloys, the melt flows out of the tundish and enters into the roll-shoe gap, and then solidifies due to the cooling effect caused by the water-cooled roll and the shoe. The roll is cooled by circulating water, which offers a high cooling rate during solidification. Many studies have shown that the cooling rate has an important effect on the as-cast microstructure of alloys during solidification. The average size of grains [5], primary and/or secondary dendrite arm spacing [6], morphology and distribution of secondary phases [7], and solid solubility of alloying element [8] are proved to be affected by cooling rate. Therefore, it is necessary to determine the temperature variation of the melt, especially the influence of rotating speed and cooling water velocity on the cooling rate. Orthogonal experiments were carried out to measure the melt temperature varied with the rotating speed or the cooling water velocity [9,10]. However, it is difficult to measure the melt temperature in the roll-shoe gap during the continuous rheo-extrusion process. Therefore, it is important to establish a cooling rate calculation model for the molten metal. The distribution of temperature field, solidification rate and cooling rate in the solidification process mainly depend on the interface heat transfer coefficient between the metal and mold [11,12,13,14]. Due to the complexity of interface morphology and the uncertainty of the heat transfer mechanism in the gap, there is no effective method for quantitatively describing the interface heat transfer coefficient. Cheung et al. [15] proposed a method to establish the expression of the metal/mold interface heat transfer coefficient of the rotary continuous caster. However, for the continuous rheo-extrusion process, the influence of the heat transfer capacity of the roll and the cooling system in the roll cannot be ignored.

Al–Mg alloys (5XXX aluminum alloys) are used to be welding wires [16,17]. Compared with pure aluminum and 6XXX aluminum alloys, due to the limitation of deformation capability, there are problems for 5XXX aluminum alloys with a high Mg content prepared by rheo-extrusion, such as low product efficiency and poor quality. Therefore, it is necessary to explore the preparation process, thus optimizing the property of alloys. In this work, a calculation model for the cooling rate of aluminum alloy melts during continuous rheo-extrusion based on the theory of heat transfer is developed. Combined with the CALPHAD calculation, the melt temperature during solidification in the continuous rheo-extrusion process is calculated. The melt temperature of an Al–6Mg alloy is estimated via this model. The aim of this work is to realize the continuous rheo-extrusion process optimization and to provide a reference for simulation research.

## 2. Calculation Model and Governing Equations

The schematic physical model of the continuous rheo-extrusion process is shown in Figure 1. Solidification is the main process of transferring heat from the high-temperature melt to the roll and shoe, including the process of liquid–solid transformation. In order to simplify the calculation, the following basic hypotheses are proposed as follows:The process is a continuous and stable process. The impact, leakage and flow undulation of melt at the initial and finishing stages are neglected.The melt is incompressible viscoplastic material, and the volume constancy is satisfied.The temperature of the roll and shoe surfaces contacting to melt is considered as the constant.The heat transfer by the water in the inlet and outlet pipes is ignored.The temperature of the sections where the cooling water pipe contacting with the roll is considered to be identical.The flow velocity in the cooling water pipes is uniform, and the velocity of water is regarded as a constant value at the same geometric position.

During the solidification process of melt in the roll-shoe gap, there are two parts of heat that will mainly be transferred away by the roll, the release of latent heat and the heat comes from the decrease in the melt temperature. The energy relation equation is expressed as
(1)$$Q_{\mathrm{mr}}=Q_{m}+Q_{s}$$
where $Q_{m}$ is the released heat due to the decrease in melt temperature. It is well known that, in unit time, when the melt temperature decreases, the calculation formula of heat emission can be given by
(2)$$Q_{m}=c_{m}\rho_{m}l_{m}^{2}\left(T_{m0}-T_{m}\right)$$
where $c_{m}$ and $\rho_{m}$ are the specific heat capacity and density of melt, respectively, $l_{m}$ is the length of the cross-section of the roll-shoe gap, $T_{m0}$ is the initial temperature of melt that is equal to the pouring temperature, and $T_{m}$ is the actual melt temperature in the roll-shoe gap. The classic expression of the release of latent heat between liquidus and solidus temperatures is expressed as follows:(3)$$Q_{s}=\rho_{m}l_{m}^{2}L_{m}f_{s}$$
where $L_{m}$ is the latent heat and $f_{s}\;$ is the local solid fraction.

The melt heat in the roll-shoe gap is mainly conducted by convective heat $\;\left(Q_{\mathrm{mr}}\right)$ exchange at the melt/roll interface, which can be given by [18] (pp. 7–9)
(4)$$Q_{\mathrm{mr}}=4h_{\mathrm{mr}}l_{m}\tau \left(T_{m}-T_{r2}\right)$$
where $h_{\mathrm{mr}}$ is the heat transfer coefficient acting at the melt/roll interface, and *τ* is the heat conduction time.

By substituting Equations (2)–(4) into Equation (1), the calculation formula of melt temperature is obtained as follows: (5)$$T_{m}=\left(4h_{\mathrm{mr}}\tau T_{r2}+c_{m}\rho_{m}l_{m}T_{m0}+\rho_{m}l_{m}L_{m}f_{s}\right)/\left(4h_{\mathrm{mr}}\tau +c_{m}\rho_{m}l_{m}\right)$$

Based on the previous work, the expression of $h_{\mathrm{mr}}$ calculated as follows [15]:(6)$$h_{\mathrm{mr}}=\alpha \tau^{-m}$$
where α and m are constants. Therefore, it is necessary to determine the temperature of roll at the melt/roll interface.

The internal heat transfer process of the roll can be simplified as a cylinder heat transfer model, as shown in Figure 2.

It is considered that the heat conduction along the axial direction is negligible, and the temperature only changes along the radial direction. In addition, the temperature of roll and shoe surface contacting to melt is considered to be a constant. Thus, the heat conduction inside the roll is described by [18] (pp. 136–154)
(7)$$Q_{r}=\frac{2\pi \lambda_{r}l_{3}\left(T_{r1}-T_{r2}\right)}{\ln\left(R_{2}/R_{1}\right)}$$ where $\lambda_{r}$ is the heat conductivity of the low-carbon steel roll, $l_{3}$ is the length that is shown in Figure 3, $T_{r1}$ is the roll temperature at the cooling water/roll interface, $T_{r2}$ is the roll temperature at the melt/roll interface, $R_{1}$ is the roll radius at the cooling water pipes position, and $R_{2}$ is the roll outside radius.

Heat transfer between the melt and roll is the key part of the melt treatment. According to Equation (4), the heat transfer during solidification is mainly determined by the two key factors, i.e., the heat transfer coefficient and the roll temperature at the melt/roll interface. In addition, the heat transfer coefficient at the melt/roll interface is also significantly affected by the temperature of the roll at the melt/roll interface. As shown in Figure 3, it is very difficult to directly detect the roll surface temperature in practical applications owing to the complex structure of the equipment. The cooling water system in the roll determines the roll temperature field. By measuring the temperature before and after the cooling water flowed through the roll and the flow rate, the energy taken away by cooling water can be calculated. Thus, the roll temperature can be estimated by combining the energy equation (Equation (1)) with the heat conduction law in the roll (Equation (7)). According to Equation (7), the expression of the roll surface temperature can be written as
(8)$$T_{r2}=T_{r1}-\frac{Q_{r}\ln\left(R_{2}/R_{1}\right)}{-2\pi \lambda_{r}l_{3}}$$

According to the principle of energy conservation, the heat $\left(Q_{w}\right)$ taken away by the cooling water is equal to the convective heat $\left(Q_{\mathrm{wr}}\right)$ at the cooling water/roll interface. They also equal the conduction heat $\left(Q_{r}\right)$ inside the roll and the convective heat $\left(Q_{\mathrm{mr}}\right)$ at the melt/roll interface. Thus, there is an equivalence relation as follows: (9)$$Q_{w}=Q_{\mathrm{wr}}=Q_{r}=Q_{\mathrm{mr}}$$

$Q_{w}$ can be calculated as [18] (pp. 523–525)
(10)$$Q_{w}=c_{w}\rho_{w}u_{w}\pi {\left(\frac{d_{1}}{2}\right)}^{2}\left(T_{w2}-T_{w1}\right)$$
where $c_{w}$ is the specific heat capacity of cooling water, $\rho_{w}$ is the density of the cooling water, $u_{w}$ is the velocity of cooling water in the inlet pipe, $d_{1}$ is the diameter of the cooling water inlet pipe, $T_{w1}$ is the inlet temperature of cooling water, and $T_{w2}$ is the outlet temperature of cooling water.

According to Newton’s law of cooling, $Q_{\mathrm{wr}}$ can be expressed as follows [18] (pp.112–115):(11)$$Q_{\mathrm{wr}}=\sum h_{\mathrm{wr}}A_{\mathrm{wr}}\left(T_{\mathrm{rl}}-T_{w}\right)$$
where $h_{\mathrm{wr}}$ is the heat transfer coefficient acting at the cooling water/roll interface, $A_{\mathrm{wr}}$ is the area of contact surface at the cooling water/roll interface, $T_{w}$ is the temperature of the cooling water at the cooling water/roll interface, and $T_{w}$ is given by
(12)$$T_{w}=\frac{T_{\mathrm{wl}}+T_{w2}}{2}$$

Based on Equation (11), the heat transfer coefficient acting at the cooling water/roll interface (*h*_wr_) should be determined firstly, and then the convective heat $\left(Q_{\mathrm{wr}}\right)\;$ at the cooling water/roll interface can be calculated. To simplify the calculation, the heat transfer between the water in inlet pipes and the roll is ignored. In addition, it is assumed that the flow velocity of cooling water in inlets pipes is a constant, and the flow velocity of cooling water in the distributed pipes with the same special positions is identical. It can be seen that the convective heat $Q_{\mathrm{wr}}$ at the cooling water/roll interface can be divided into two parts. One part is the heat taken away by the cooling water in the pipes parallel to the axis of roll, and the other part is the heat taken away by the cooling water in the pipes vertical to the axis of the roll. Therefore, the convective heat $Q_{\mathrm{wr}}$ in this system is expressed as
(13)$$Q_{\mathrm{wr}}=\sum h_{\mathrm{wr}}A_{\mathrm{wr}}\left(T_{\mathrm{rl}}-T_{w}\right)=\left(h_{\mathrm{wr}1}A_{\mathrm{wr}1}+h_{\mathrm{wr}2}A_{\mathrm{wr}2}\right)\left(T_{\mathrm{rl}}-T_{w}\right)$$
where $h_{\mathrm{wr}1}$, $A_{\mathrm{wr}1}$ and $h_{\mathrm{wr}2}$, $A_{\mathrm{wr}2}$ are the convective heat transfer coefficient and the interface area corresponding to the pipes parallel and vertical to the axis of roll, respectively. The expression of convective heat transfer coefficient is given by [18] (pp. 436–441)
(14)$$h=\frac{\lambda }{d}Nu$$
where d is the diameter of the cooling water pipe. The velocity of cooling water in the inlet pipe, the velocity of cooling water in the pipes parallel and vertical to the axis of the roll are $u_{w}$, $u_{w1}$ and $u_{w2}$, respectively. The equal volume of cooling water can be described as follows:(15)$$\pi {\left(\frac{d_{1}}{2}\right)}^{2}u_{w}=8\frac{l_{2}-l_{1}}{2}\frac{d_{2}}{2}u_{w2}=8\frac{1}{2}\pi {\left(\frac{d_{2}}{2}\right)}^{2}u_{w1}$$
where $d_{1}$ is the diameter of the cooling water inlet pipe, $d_{2}$ is the diameter of the cooling water pipe
parallel to the axis of the roll, *l_1_* and *l_2_* are the lengths that are shown in Figure 3.

The well-known calculation formula of Reynolds number (*R_e_*) is described as follows [18] (pp. 434–441): (16)$$Re=\frac{ud}{\upsilon }$$
where *u* is the viscosity of cooling water, and υ is the motion viscosity of cooling water. The Re value is an empirical value to judge the turbulent or laminar flow in a smooth tube. In this case, the critical Re value is 10^4^ [19] (pp. 246–249). The equivalent diameters of the cooling water pipe parallel and vertical to the axis of the roll are $d_{3}$ and $d_{4}$, respectively. In addition, the value of υ is taken as the water motion viscosity at the average fluid bulk temperature $\left(T_{w}\right)$, thus the critical viscosities of $u_{w1}$ and $u_{w2}$ are 5.1538 m/s and 6 m/s, respectively. Considering the actual situation, the viscosity of cooling water in the inlet pipe is below 5 m/s, therefore, only the heat transfer in the laminar flow system is studied.

A simper relation proposed by Sieder and Tate for laminar heat transfer in tubes is shown as Equation (17), which can be used to calculate the Nu corresponding to the pipes parallel to the axis of the roll [19] (pp. 252).
(17)$$Nu_{1}=1.86{\left(\frac{RePr}{1/d}\right)}^{1/3}{\left(\frac{\mu_{w}}{\mu_{r}}\right)}^{0.14}$$
where Prandtl number (*Pr*) [18] (pp. 407–409) is defined as the ratio of the kinematic viscosity (υ) to the thermal diffusivity (α), and the expression is
(18)$$Pr=\frac{\upsilon }{\alpha }=\frac{\mu c}{\lambda }$$

The pipes parallel to the axis of the roll are circular, and a correction factor $C_{r}=1+10.3{\left(\frac{d}{R}\right)}^{3}$ is introduced to Equation (17), thus the $Nu_{2}$ can be written as [19] (pp. 248–249)
(19)$$Nu_{2}=\left[1+10.3{\left(\frac{d}{R}\right)}^{3}\right]1.86{\left(\frac{RePr}{1/d}\right)}^{1/3}{\left(\frac{\mu_{w}}{\mu_{r}}\right)}^{0.14}$$

Substituting Equations (17) and (19) into Equation (14), respectively:(20)$$h_{\mathrm{wr}1}=\frac{\lambda_{w}}{d_{3}}Nu_{1}=\frac{\lambda_{w}}{d_{3}}1.86{\left(\frac{0.16u_{w}d_{1}}{\upsilon_{w}}\cdot \frac{Pr}{1/d_{3}}\right)}^{1/3}{\left(\frac{\mu_{w}}{\mu_{r}}\right)}^{0.14}$$
(21)$$h_{\mathrm{wr}2}=\frac{\lambda_{w}}{d_{4}}Nu_{2}=\frac{\lambda_{w}}{d_{4}}\left[1+10.3{\left(\frac{d_{2}}{R}\right)}^{3}\right]1.86{\left(\frac{0.04\pi u_{w}d_{2}}{\upsilon_{w}}\cdot \frac{Pr}{1/d_{4}}\right)}^{1/3}{\left(\frac{\mu_{w}}{\mu_{r}}\right)}^{0.14}$$

## 3. Results and Discussion

Figure 4 shows the curves of convective heat transfer coefficients varied with the velocity of cooling water. The fluid properties are calculated at the average temperature of cooling water. When $T_{w1}$ and $T_{w2}$ are taken as 15 °C and 25 °C, respectively, then $T_{w}$ is 20 °C. The physical property constants of water at 20 °C are listed in Table 1. It can be seen that with the increase in cooling water velocity, the convective heat transfer coefficients increase gradually, and the heat transfer coefficients in laminar flow are below 1200 W/(m^2^·K).

The temperature of roll at the cooling water/roll interface ($T_{r1}$) can be obtained by combining Equations (9)–(11):(22)$$T_{r1}=\frac{c_{w}\rho_{w}u_{w}\pi {\left(\frac{d_{1}}{2}\right)}^{2}\left(T_{w2}-T_{w1}\right)}{\sum h_{\mathrm{wr}}A_{\mathrm{wr}}}+\frac{T_{\mathrm{wl}}+T_{w2}}{2}$$

The curve of $T_{r1}$ varied with cooling water velocity is shown in Figure 5. It can be seen that the roll temperature at the cooling water/roll interface increases with the increase in cooling water velocity and increases when the velocity of cooling water is increased. When the temperature increment of cooling water is fixed, the greater the cooling water velocity is, the more heat the cooling water takes away in unit time. Moreover, the heat transfer process is associated with the thermal conductivity of roll [20]. Therefore, only when enough heat is transferred in the roll, can the heat transfer increment caused by the increase of cooling water velocity be satisfied. Therefore, there is a large temperature gradient in the roll to meet the condition, and the temperature of the roll at the melt/roll interface will also increase with the increase in cooling water velocity. 

By substituting Equation (22) into Equation (10), the calculation formula of the roll temperature at melt/roll interface is given as follows:(23)$$T_{r2}=c_{w}\rho_{w}u_{w}\pi {\left(\frac{d_{1}}{2}\right)}^{2}\left(T_{w2}-T_{w1}\right)\left[\frac{1}{\sum h_{\mathrm{wr}}A_{\mathrm{wr}}}-\frac{\ln\left(R_{2}/R_{1}\right)}{-2\pi \lambda_{r}l}\right]+\frac{T_{\mathrm{wl}}+T_{w2}}{2}$$

The curve of $T_{r2}$ varied with cooling water velocity is shown in Figure 6. The rule shown by the curve is that the roll temperature at the cooling water/roll interface increases with the increase of cooling water velocity. As stated above, there is a large temperature gradient in the roll. Although the cooling water is laminar, and the heat transfer efficiency of the contact interface between the cooling water and the extrusion roller is low. There is a high-temperature gradient in the roll to achieve the required heat transfer efficiency.

Hence, the melt temperature during the continuous rheo-extrusion process can be established by combining Equations (5), (6) and (23). In this case, *a* and *m* in Equation (6) are constants, and the value of *m* is less than 0.5. τ is the time of heat transfer between the melt and roll-shoe gap, which mainly is related to the convective heat transfer and the rotational speed of the roll. Figure 7 shows the temperature variation of the Al–6Mg melt in the roll-shoe gap during continuous rheo-extrusion. According to CALPHAD calculation, the density of Al–6Mg (wt.%) alloy melt is 2.31 × 10^6^ g·m^−3^ and the specific heat is 1.1594 J/(g·K) at 700 °C. Based on the above calculation results, for the Al–6Mg alloy prepared by continuous rheo-extrusion, when the cooling water flow rate in the cooling water pipe is 1.5 m/s and the melt is poured into the roll-shoe gap at 800 °C, the melt temperature is rapidly dropped to the liquidus temperature within 1.7 s and completely solidified in 7.5 s. Temperature variation of the Al–6Mg melt in the roll-shoe gap with increasing time during continuous rheo-extrusion is shown in Figure 7. In this cooling condition, the average cooling rate is about 10.3 K/s.

During continuous rheo-extrusion, it is very difficult to accurately measure the actual melt temperature in the roll-shoe gap due to the complexity of the equipment. After establishing the calculation model, the accuracy of the calculation is indirectly verified by other methods. The solidification microstructure of the Al–6Mg alloy in the roll-shoe gap is obtained by shutdown sampling, as shown in Figure 8a. In addition, the solidification microstructures of Al–6Mg alloys with various cooling rates (5 K/s, 10 K/s, and 20 K/s) also are achieved by traditional metal mold casting, as shown in Figure 8b–d. The wall thickness of the mold is different, and the melt temperature is measured and recorded by using a self-made temperature acquisition system. By comparing the average grain size and secondary dendrite arm spacing, the microstructure of the alloy prepared by continuous rheo-extrusion is similar to the solidification microstructure of 10 K/s. This indicates that the calculated cooling rate (10.3 K/s) is within the credible range.

## 4. Conclusions

This work proposes an estimation methodology to calculate the melt temperature during solidification in the continuous rheo-extrusion process via the basis theory of heat transfer and the CALPHAD calculation. The computational models of heat transfer coefficients acting at the cooling water/roll interface and the melt/roll interface are established. The expression of the relationship between the temperature of the roll at the melt/roll interface and the viscosity of the cooling water is determined. Using this method, the cooling rate of Al–6Mg (wt.%) alloy melt prepared through continuous rheo-extrusion in the equipment is calculated to be ~10.3 K/s.

## Figures and Tables

**Figure 1 materials-14-05684-f001:**
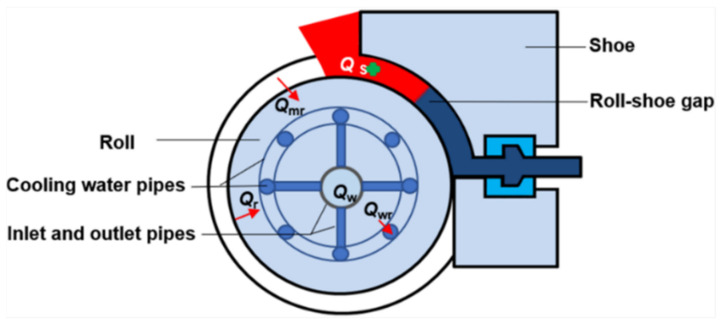
Schematic physical model of the continuous rheo-extrusion process.

**Figure 2 materials-14-05684-f002:**
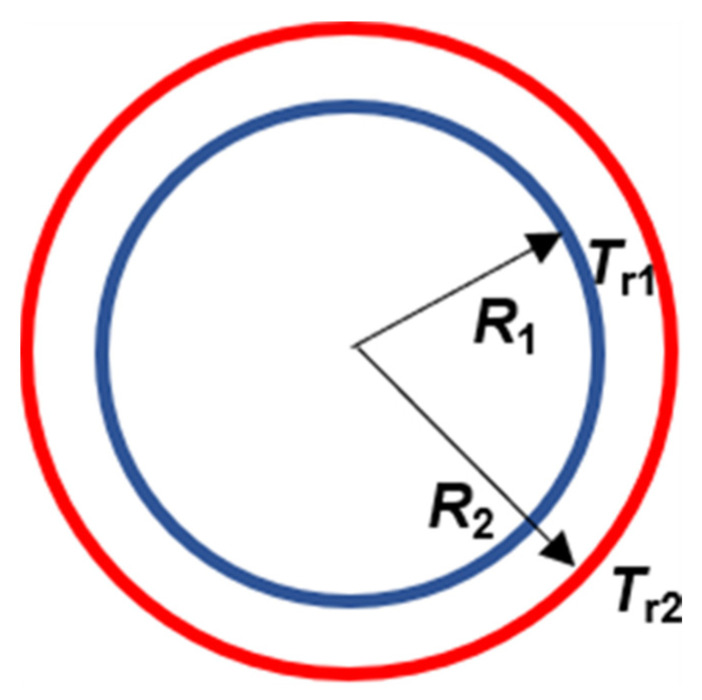
Diagram of the simplified model of the cylinder heat transfer model.

**Figure 3 materials-14-05684-f003:**
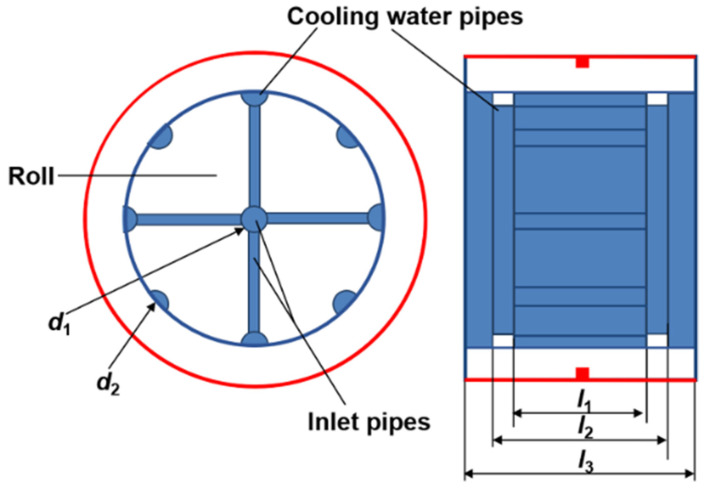
Distribution diagram of the cooling water system in the roll.

**Figure 4 materials-14-05684-f004:**
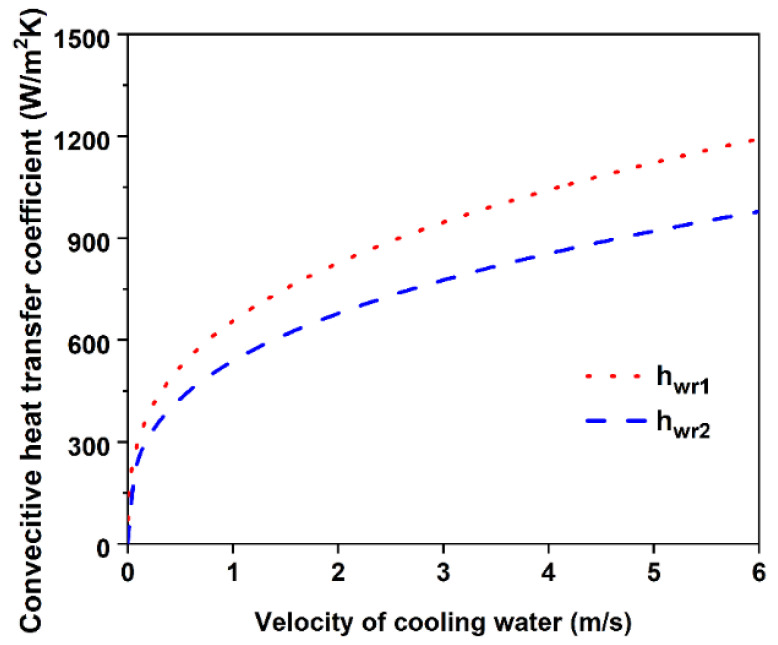
Curves of convective heat transfer coefficients varied with velocity of cooling water.

**Figure 5 materials-14-05684-f005:**
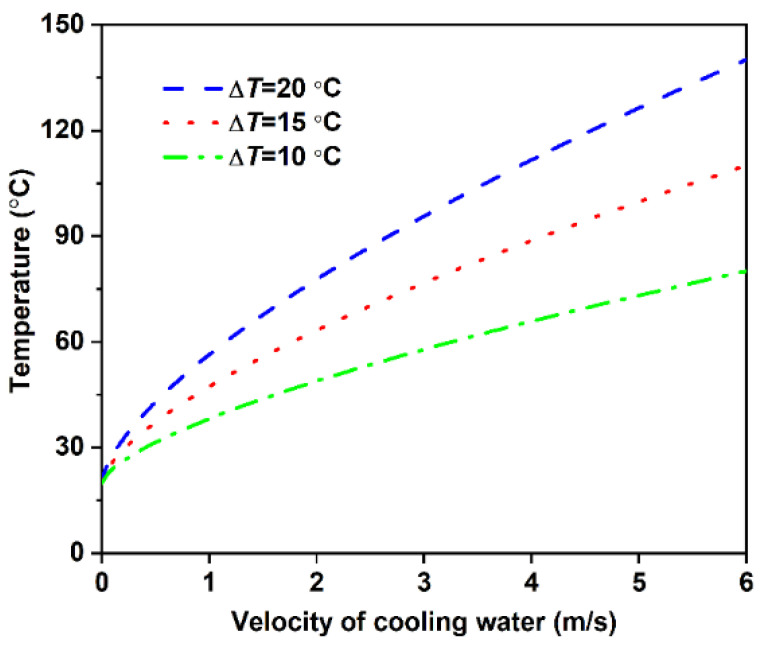
Roll temperature at the cooling water/roll interface as a function of velocity of cooling water.

**Figure 6 materials-14-05684-f006:**
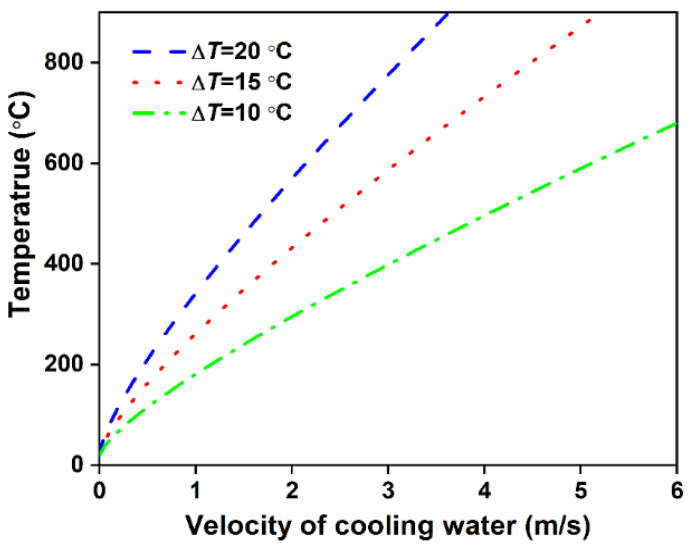
Relationship between the roll temperature at the melt/roll interface and cooling water velocity.

**Figure 7 materials-14-05684-f007:**
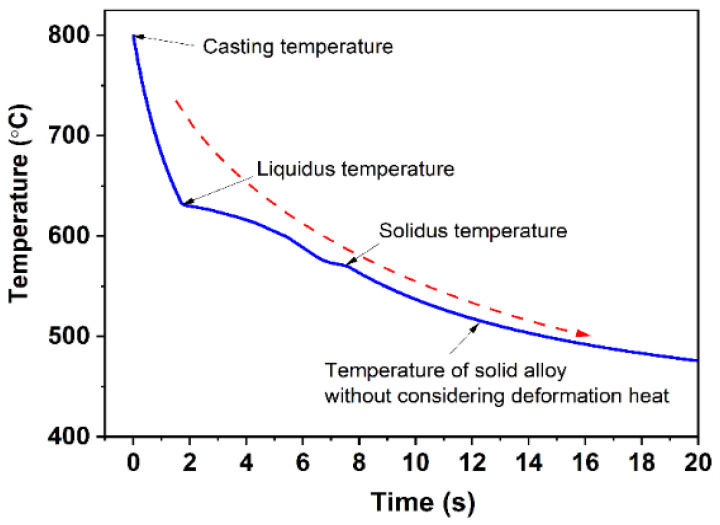
Temperature variation of the Al–6Mg melt in the roll-shoe gap with increasing time during continuous rheo-extrusion.

**Figure 8 materials-14-05684-f008:**
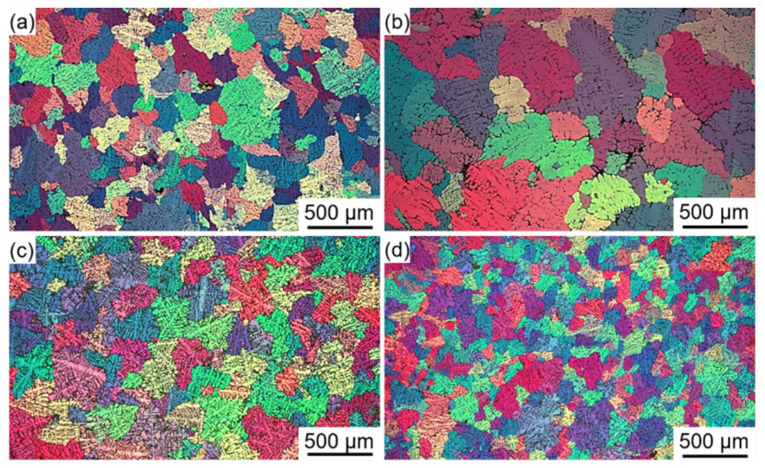
(**a**) Solidification microstructure of the Al–6Mg alloy in the roll-shoe gap obtained by shutdown sampling; (**b**–**d**) solidification microstructures of Al–6Mg alloys with different cooling rates (5 K/s, 10 K/s, and 20 K/s) obtained by traditional metal mold casting.

**Table 1 materials-14-05684-t001:** Physical parameters of cooling water at 20 °C [19] (p. 563).

$\mathrm{Density}\;\rho_{W}{,}_{\;}\mathrm{kg}/m^{3}$	9.982 × 10^2^
Constant pressure specific heat capacity, $c_{w}$, J/(kg·K)	4.183 × 10^3^
Motion viscosity, $\upsilon_{w}$, m^2^/s	1.006 × 10^−6^
Thermal conductivity, $\lambda_{w}$, W/(m·K)	0.599
Viscosity, $\mu_{w}$, Pa·s	1.004 × 10^−9^

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
