# Peer review of "A Calculation Model for Cooling Rate of Aluminum Alloy Melts during Continuous Rheo-Extrusion"

_materials, 2021, doi:10.3390/ma14195684_

Round 1
Reviewer 1 Report
The manuscript "A calculation model for cooling rate of aluminum alloy melts during continuous rheo-extrusion" is interesting and well-structured.
The authors described the steps of the model carefully, and the related images are helpful for the reader to understand. The English language is also well-written. The conclusion section is concise but clear.
I only note that Qm, used and described in lines 77-85, is missing from the Nomenclature table on page 2. Maybe it was intended, or it is just a slip because I noticed that the other melt-related parameters are listed in the Nomenclature table.
Reviewer 2 Report
This paper is a Fermi estimate of the convective heat transfer coefficient.
The Reynolds number was defined but the Prandtl number wasn't.
The Nusselt number is dependent on the ratio of the Reynold and Prandtl number and is characteristic of the system. These should be double checked with a experimental results to verify the accuracy of the Nusselt number calculations.
Additionally, FEM simulations could have been done to verify that this approach works - since fluid velocity will be known or can be measured from the inlet pipe, and the rotation of the rolls can be estimated from the motor RPM.
I don't feel that there is any new knowledge contained in this piece of work. Sorry.
Reviewer 3 Report
The authors did a good job, with a direct impact on production efficiency and reducing scrap. However, I have a few comments:
1. line 33 - 35. You state: " Many studies have 33 shown that the cooling rate has an important effect on ... ." Explain and state how cooling affects the microstructure
2. line 61 - 71. Which happens when the conditions you specify are violated? Please indicate the marginal options for violating the terms.
3. Is your calculation verified in industrial practice or by means of simulation programs for casting?
Reviewer 4 Report
Review
The article is devoted to the calculation of the cooling rate of an aluminum alloy during rheo-extrusion. For this, the well-known provisions of the basic theory of heat transfer were applied.
Notes on the article:
Why did you use this particular grade of aluminum alloy (Al - 6Mg)? Why did not the calculation be carried out for aluminum alloys of different composition (system), purpose, etc. It was necessary to take at least 5-10 marks. Indeed, one alloy cannot judge the effectiveness of this calculation.
During the calculation, the technological parameters of the process did not vary to the proper extent. Such as the flow rate of cooling water, the temperature of the poured melt, etc.
It seems that the authors of the article were solving some very narrow point problem, but nothing more.
Based on the above, the conclusions are insufficiently substantiated. The calculation method requires more serious approbation on other grades of aluminum alloys with a wide variation of the technological parameters of the process.
Literary sources are clearly insufficient. At the same time, out of 18 sources, 13 have a term older than 10-20 years.
The article has a pronounced applied character.
From a scientific point of view, the value of the article is not high enough. A very narrow study.
The article requires revision, it is necessary to take into account all the above comments.
Round 2
Reviewer 4 Report
The authors responded to all comments of the reviewer. The article can be recommended for publication.